# Control Strategy of 1 kV Hybrid Active Power Filter for Mining Applications

**Dawid Buła \***, **Jarosław Michalak**, **Marcin Zygmanowski**, **Tomasz Adrikowski, Grzegorz Jarek and Michał Jeleń**

Faculty of Electrical Engineering, Silesian University of Technology, 44-100 Gliwice, Poland;
jaroslaw.michalak@polsl.pl (J.M.); marcin.zygmanowski@polsl.pl (M.Z.); tomasz.adrikowski@polsl.pl (T.A.);
grzegorz.jarek@polsl.pl (G.J.); michal.jelen@polsl.pl (M.J.)
**\*** Correspondence: dawid.bula@polsl.pl

**Abstract:** The paper presents a shunt hybrid active power filter for 1 kV mining applications in configuration with a single-tuned passive filter. The focus is on the method of controlling the filter, with particular emphasis on the influence of network voltage distortion and time delays in control on the effectiveness of harmonic reduction, which is one of the most important aspect of power quality. The low-power loss configuration of a hybrid filter with SiC transistors is shown, as well as the control algorithm which limits the influence of voltage distortion. Theoretical considerations are verified by results obtained from simulations and tests of the hybrid active power filter prototype.

**Keywords:** power quality; hybrid active power filters; harmonic reduction; control algorithms; time delay; voltage distortion





## 1. Introduction

Problems related to the power quality (PQ) in distribution grids have been known for many years. However, recently they have become increasingly important. The reason for this is, on the one hand, the dependence of modern society on electricity, and on the other, the increase in the number of loads affecting power quality. The consequence of continuous technological progress in the field of electronics is the increase in the number of power electronic converter-based appliances connected to the grid. Converters in drive systems with AC motors, switching power supplies, drives with brushless DC motors, battery chargers or energy-saving lighting, despite their advantages, deteriorate the power quality. Moreover, the number of low-power appliances using power electronic systems in households is constantly growing, and the number of the power electronic-based drive systems in the industry is increasing. For example, according to [1], the size of the global market of variable frequency drives was valued at 22.5 billion USD in 2019 and is expected to grow at a compound annual growth rate of 6.5% from 2020 to 2027. The use of modern solutions is associated with the deterioration of the PQ. It causes greater variability of disturbances in the distribution grid and limits the possibility of reducing these disturbances at the source side. In industry, an increasing number of technological processes are sensitive to changes in power supply parameters and even slight deviations can cause large economic losses. From the economical point of view, effects of low power quality can be divided into four categories [2,3]:

- partial or total loss of one or more processes (e.g., loss of control due to a voltage dip);
- poor long-term performance or poor product quality (e.g., employee fatigue due to flicker);
- cost increase due to reduction of equipment life leading to premature failure (e.g., overheating of transformers due to harmonics);
- increased power losses resulting from distorted voltages and currents.

According to [2], the impact of harmonics on the costs related to power quality is only 5.4% of the total PQ cost, but it is still a huge sum. The authors of the report for 2005–2006 [3] estimate that the losses in the European Union related to the quality of electricity amounted to approximately 150 billion EUR at that time.

The applicable standards and recommendations [4–6] impose on manufacturers a requirement to limit the higher harmonic currents consumed by the manufactured devices. As this does not solve the problem, newer ways to reduce higher harmonics are constantly being developed. The classic solution, which is the use of the resonant passive filters (PF), is still the cheapest. However, much better results are obtained by using active power filters (APF) or hybrid active power filters (HAPF). Therefore, a large number of publications focus on new configurations and methods of controlling APF and HAPF.

The first works on hybrid active power filters appeared in the late 1980s and early 1990s. The combination of a passive and an active power filter in one application was first described in [7] (based on [8]). Subsequent described applications concerned the combination of a series active power filter with a parallel passive filter [9] and a parallel active filter with a parallel passive filter [10]. Historically, mention should be made of [11], in which a combination of earlier solutions was shown. Since then, there have been publications on increasingly newer system solutions and methods of controlling HAPF. Therefore, two main directions of research can be distinguished. The first, concerning new system solutions, focuses on a new configuration of the hybrid active power filter or a modification of the existing one in order to improve certain properties of the system. Modifications most often come down to changing the configuration of the passive filter [12–16], the way of its connection with the active part [17–19] or the use of new converter structures [20,21]. The second research direction is the search for new control algorithms. Apart from trying to apply one of the power theories [22–25], the authors use techniques such as: artificial neural networks [18,19], fuzzy logic [26–29], adaptive filters [28,30] or other methods [31–34]. The key issues here are also the methods of controlling the converter dc voltage [35–37] or the elimination of the influence of the supply voltage distortion on the operation of the HAPF [38,39]. Originally, the hybrid active power filters were used in the medium and the high power systems [40–43], where the use of passive filters did not give the expected results, and the use of active power filters was impossible (e.g., due to high voltages) or unprofitable for economic reasons. In recent years, however, the amount of low and medium power non-linear loads, which are the source of higher harmonics in the grid currents and voltages, has been increased. Therefore, interest in the HAPF operating with lower power loads and reduced passive part has increased [44–48].

This work concerns the hybrid active power filter applied in mining excavations with high concentrations of methane. It requires the HAPF to be mounted inside of the explosion-proof housing, in accordance with the guidelines contained in ATEX standards [49–51]. The device belongs to the group I [49], which concerns devices intended for use in the methane mining. The HAPF is classified in the category M2 because the rated voltage and current exceed defined limits [49,52]. Devices from this category have to be automatically disconnected from the power supply when the permissible methane concentration is exceeded (the alarm threshold is set at 1.5 ÷ 2%). Therefore, it has been covered by a single explosion protection measure by placing it in a flameproof enclosure with a high level of protection "db" [50] and a high degree of protection Mb (the device will not become a source of ignition of an explosive mixture between its appearance and the automatic shutdown of the device). The ATEX marking of the equipment is as follows I M2 Ex db I Mb. Placing the device in an explosion-proof housing is associated with limited possibilities of heat dissipation from the inside to the outside of the casing, in addition, high ambient temperature of up to approx. 40 °C and very high humidity significantly deteriorate the cooling properties of the casing. Therefore, it is extremely important to minimize the power losses dissipation inside of the housing.

The use of the APF for HAPF filters in the mining electrical grid may contribute to the reduction of conducted and radiated EMC disturbances by eliminating harmonics in

the grid currents and voltages. Furthermore, it should reduce the number of interruptions and disturbances in the operation of atmosphere parameter monitoring systems, thus increasing the level of safety of mine workers [53,54]. The application of HAPF yields financial benefits due to lowering power losses and slowing down the aging effects of appliances e.g., motor drives [55–58].

The paper presents a shunt hybrid active filter for 1 kV mining applications in configuration with a single-tuned passive filter. Usually, mine electrical grids require using long cables, which in the case of non-linear loads causes voltage distortion [59,60]. Therefore, in addition to the appropriate topology with SiC transistors allowing high energy efficiency and low losses to be achieved, an original solution related to the control was proposed. These solutions allow to achieve high effective harmonic reduction. Among the proposed control modifications, it is worth enumerating the compensation of the influence of distortions in the network voltages and the compensation of processing delays as well as the appropriate implementation of the control algorithm. All the proposed solutions have been confirmed by the results of experimental studies.

## 2. Analyzed Hybrid Active Power Filter

The main application of the analyzed hybrid active power filter HAPF is the mining industry. The need for operation at a voltage level of 1000 V and limited heat transfer from the explosion-proof housing are the main factors in the selection of the HAPF topology. This solution operates with lower dc link voltage (which allows to reduce power losses) in comparison to active power filters and guarantees better performance compared to passive filters. Most of the currently proposed hybrid active power filter designs adopt a series connection of passive filters and a power electronic converter-based active power filter. The proposed solution is the variant with a single-tuned passive harmonic filter. Although the 5th harmonic is dominant in the electrical grids, a passive filter tuned to the 7th harmonic was used, making this filter smaller and lighter. It has to be noted that by using appropriate control of the power electronic converter in HAPF, it is also possible to ensure high harmonic attenuation rate also for 5th and 11th current harmonics, which will be thoroughly described in Section 3. The connection of the active and passive parts in HAPF enables reducing the rated power of the converter in the active power part. Moreover, the dc link voltage and the voltage class requirements of the inverter transistors and capacitors are also reduced. In the case of use of the APF for the 1 kV electrical grid, the converter dc link voltage should be higher than 1600 V. In case of use of the HAPF solution, the dc link voltage being equal to 400 V is sufficient. Figure 1 shows the connection of the proposed HAPF to the grid and idea of its operation where the $i_G$, $i_L$ and $i_F$ are the grid current, the load current and the filter current, respectively.

Figure 2 shows the detailed topology of the HAPF. There are four functional blocks in it. The first block is a security switchgear with contactors having two features. They connect the converter to the grid through start-up resistors and then connect the converter to the grid directly.

The next block in Figure 2 is the passive filter tuned to the 7th harmonic. It consists of a three-phase choke with an inductance of 4.6 mH and three capacitors with a capacitance of 45 μF. The parameters of the filter have been chosen in such a way as to reduce the fundamental reactive current of passive filter on the one hand and not to increase the size and weight of the inductor on the other.

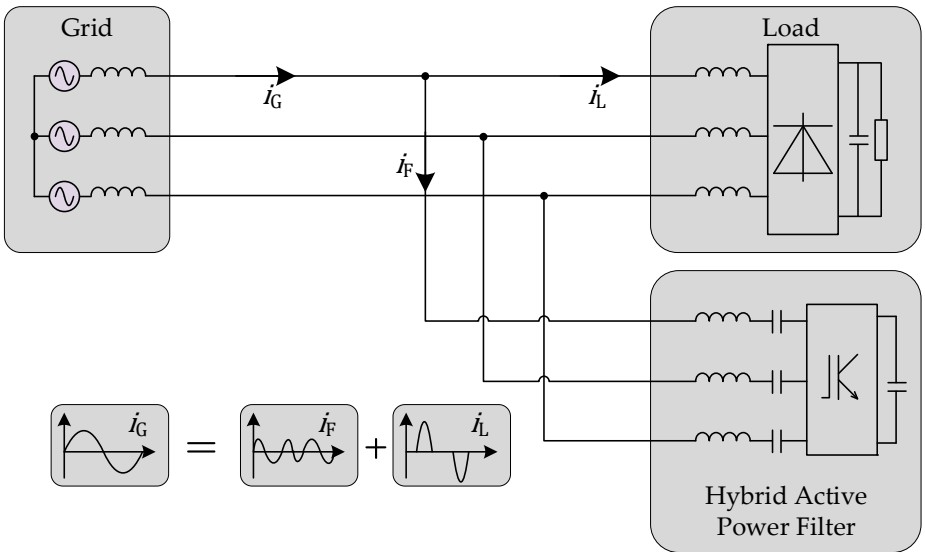

**Figure 1.** Connection of the proposed hybrid active power filter to the grid.

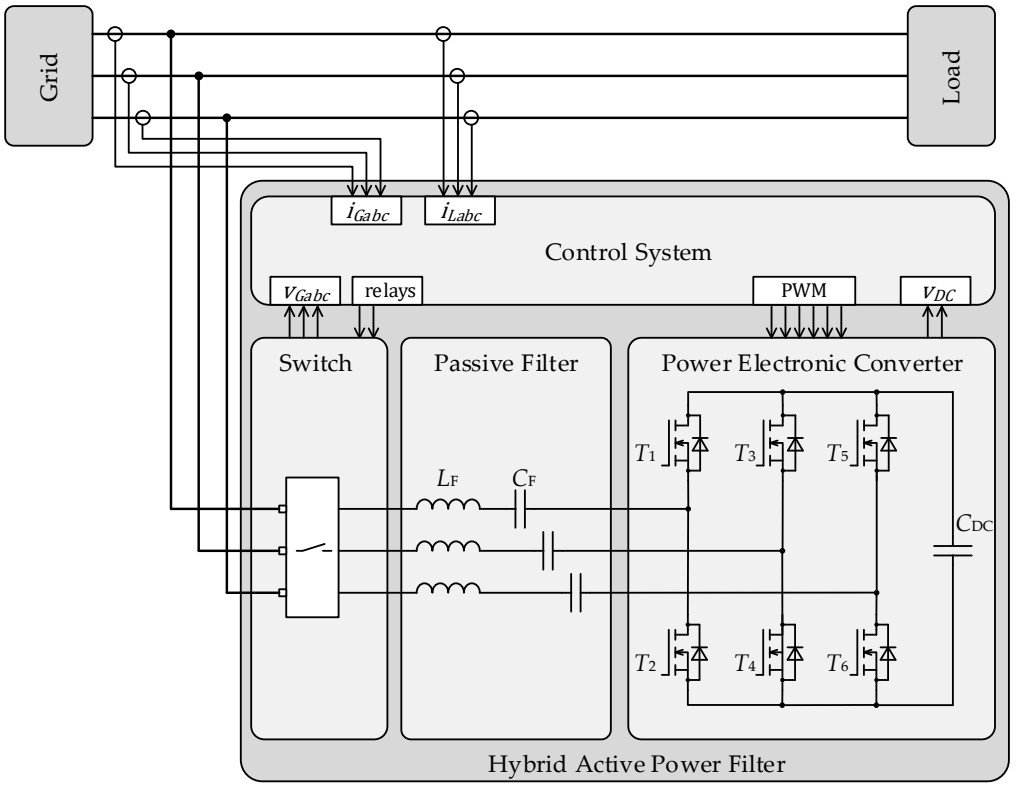

**Figure 2.** Topology of the proposed hybrid active power filter.

The last blocks inside the HAPF are the power electronic converter with its control system. The power electronic converter is based on three-phase two-level topology. Moreover, thanks to the use of the hybrid structure of the filter, the dc link voltage is relatively low, which is selected as 400 V. This enables the use of transistors with 600 V or 1200 V voltage class. This significantly reduces costs and allows for reducing power losses. Since the reduction of generated heat is an important aspect in such applications, SiC transistors with low on-state resistance have been chosen. Detailed information about the converter prototype are provided in Section 6.

### 3. Control Algorithm of the Hybrid Active Power Filters (HAPF)

The idea of the operation of the power electronic converter in a hybrid active power filter is based on increasing harmonic attenuation rate $\gamma(j\omega)$ defined as [9,10]:

$$\gamma(j\omega) = \frac{I_{Gh}(j\omega)}{I_{Lh}(j\omega)} \tag{1}$$

where: $I_{Gh}(j\omega)$ represents the grid current harmonics and $I_{Lh}(j\omega)$ represents the load current harmonics.

For the parallel passive filter PF, the harmonic attenuation rate $\gamma_1(j\omega)$ is defined as:

$$\gamma_1(j\omega) = \frac{Z_{PF}(j\omega)}{Z_{PF}(j\omega) + Z_G(j\omega)} \tag{2}$$

where: $Z_{PF}(j\omega)$, $Z_G(j\omega)$—are impedances of passive filter and the grid respectively.

The basic scheme representing the idea of the control of a HAPF is presented in Figure 3a where the power electronic converter is presented as an controlled voltage source $v_{REF}$. The control system is based on the synchronous reference frame method with feedback and feedforward control, where two control loops are used [40,61]. The first (feedback) control loop CL1 is based on the measurement of the grid currents. In CL1, higher harmonic components are filtered from the grid currents $i_G$ and the power electronic converter generates voltage that is proportional to the aforementioned current higher harmonics with gain $K$ (expressed in ohms). For the hybrid active filter with CL1 control loop and gain $K$, the harmonic attenuation rate $\gamma_2(j\omega)$ is defined as:

$$\gamma_2(j\omega) = \frac{Z_{PF}(j\omega)}{Z_{PF}(j\omega) + Z_G(j\omega) + K} \tag{3}$$

It can be observed from Figure 3b that using of CL1 control loop with gain $K$ increases the harmonic attenuation rate of the hybrid active power filter in comparison to the passive filter.

The second (feedforward) control loop, which is referred to as CL2, is based on the measurement of load current $i_L$ and generation of voltage harmonics which force selected current harmonics to flow through the passive filter PF. Using the CL2 control loop increases the harmonic attenuation rate for selected harmonics $\gamma_3(j\omega)$ as depicted in (4).

$$\gamma_3(j\omega) = \frac{Z_{PF}(j\omega) - \sum_{h=5,11,\ldots} K_h(j\omega) Z_{PF}(j\omega)}{Z_{PF}(j\omega) + Z_G(j\omega) + K} \tag{4}$$

where: $K_h(j\omega)$—represents the bandpass filtering transfer function of the CL2 control loop for a selected harmonic order $h$ [62].

The results of the use of the CL2 control loop are presented in Figure 3b as characteristic of the harmonic attenuation rate $\gamma_3$ (where 5th and 11th harmonics are taken into account).

The structure of the control system with both control loops is presented in Figure 4. For synchronization with the grid voltage $v_G$ the SOGI-FLL (second-order generalized integrator-frequency locked loop), algorithm [63] has been used. For the CL1 control loop, the transformation of the grid current $i_G$ to a coordinate system $d,q$ that rotates with fundamental frequency $\omega_1$ is used. At the next stage, the high pass filters (HPF) are used for rejecting the fundamental frequency signals.

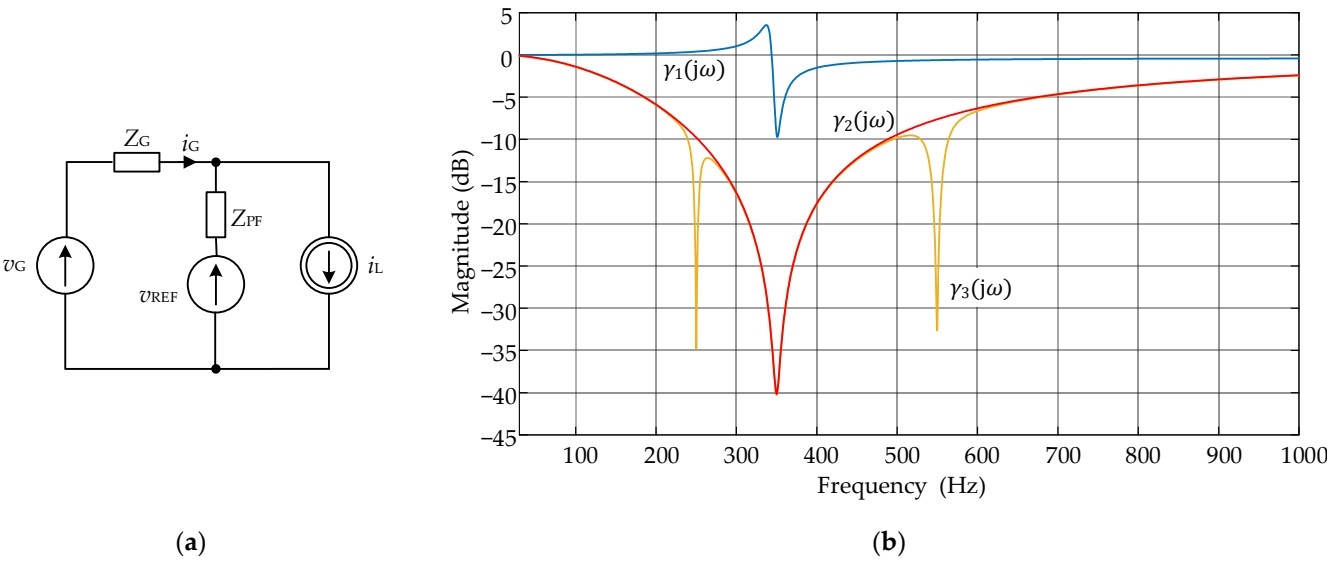

(**a**)                               (**b**)

**Figure 3.** Idea of operation of hybrid active power filter HAPF: (**a**) basic scheme of HAFP; (**b**) harmonic attenuation rate for: passive filter PF tuned for 7th harmonic ($\gamma_1$); hybrid active power filter HAPF with control loop CL1, $K = 20$ ($\gamma_2$) and hybrid active power filter HAPF with control loops CL1 and CL2, $K = 20$, CL2 for 5th and 11th harmonics ($\gamma_3$).

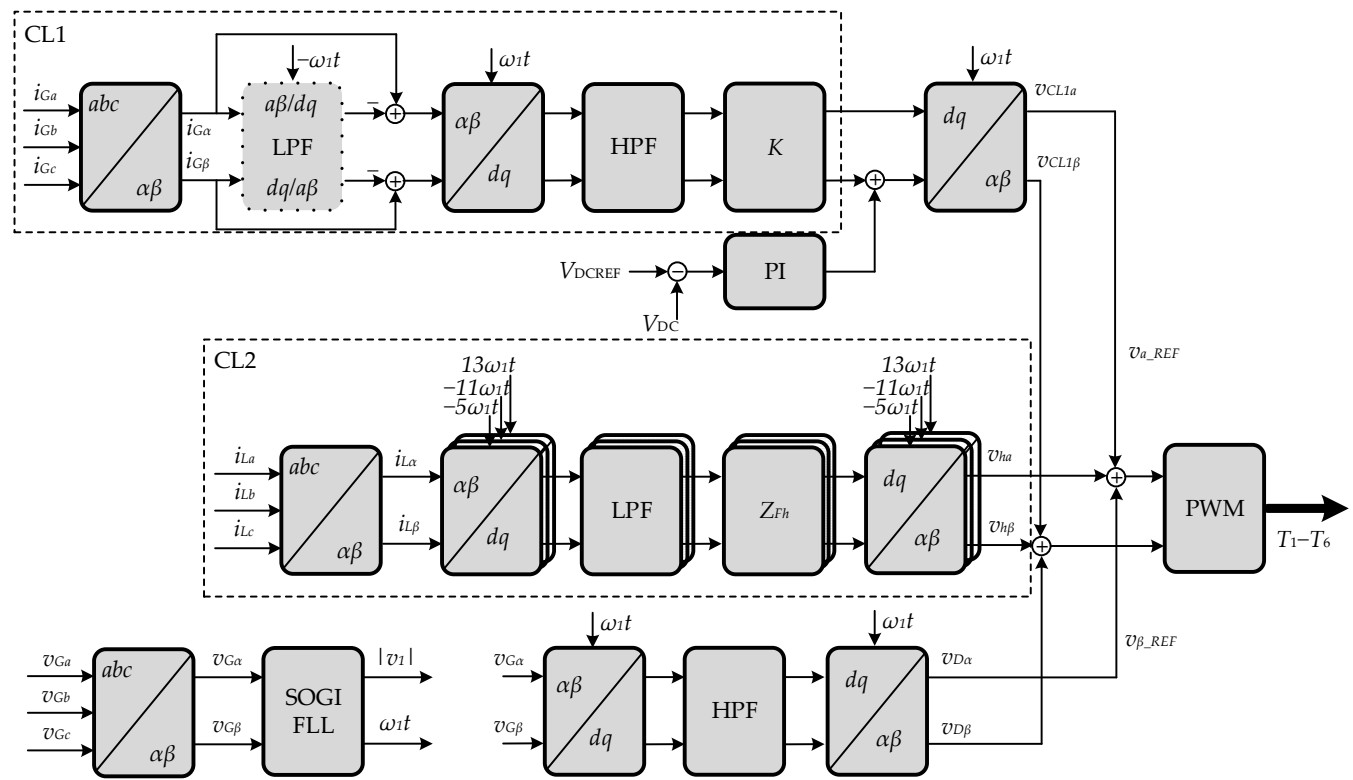

**Figure 4.** The control system of the proposed hybrid active power filter.

For better filtering effect, the transformation to a negative sequence rotating coordinate system can be used, as is presented with dotted line in Figure 4. After that, the required signals are gained with $K$ factor and transformed to stationary coordinate systems $\alpha$, $\beta$. It should be mentioned that this part of control algorithm operates as a closed loop control with the feedback and the value of the gain $K$ is limited because of the possibility of control instability, which is presented in the next section.

Typically, three-phase non-linear loads generate 5th, 7th, 11th, 13th etc. harmonics and for one of them the passive filter PF is tuned. In the CL2 control loop, which is feedforward, the load currents are transformed to the number of coordinate systems that rotate with the selected frequencies related to the harmonic order required to be reduced. The selection of such a harmonic order is done with the omission of the harmonic which the passive filter is tuned to. In the presented hybrid filter, the passive part is tuned to the 7th harmonic.

In the coordinate transformation, the sequence (positive or negative) for selected harmonic should be taken into account. Figure 4 presents CL2 for the 5th, 11th and 13th harmonics where the 5th and 11th harmonics are taken with the negative sequence and 13th is taken with the positive sequence. Additional harmonics can be added by increasing the number of blocks connected in parallel in Figure 4. For each harmonic, low pass filters (LPF) are used to filter current components for the selected frequency. After that, the impedance matrices $Z_{\mathrm{Fh}}$ are used for generation a voltage drops on passive filter impedance for the selected harmonic. This will cause the increase of the harmonic attenuation rate for the selected harmonic. After that, the output signals are transformed back to the stationary coordinate system. The CL2 control loop is feedforward open loop control where the time delay is not a critical issue, because this will not lead to operation instability. However, as will be mentioned later, these delays could cause inefficient operation of the CL2 control loop.

For the proper operation of the hybrid active power filter, two more issues are very important. The first is dc link voltage control and the second is reduction of the grid voltage harmonics influence. Due to the fact that for the fundamental harmonic the passive filter operates with the reactive power and the power electronic converter operates as a voltage source, the dc link voltage control is based on generation of fundamental $q$ component of the converter voltage (PI controller in Figure 4). This ensures proper control of the dc link voltage. The level of dc link voltage is lower than the required voltage level in grid-tied converters connected via L or LCL filters (for example the shunt active power filters or active front end converters). This is a result of a series connection of the passive filter and the power electronic converter which forms a hybrid active power filter. The lower level of voltage on the one hand ensures the reduction of power losses in power electronic converter due to the decrease of switching power losses, but on the other hand, a higher level of dc link voltage ensures the operation of HAPF with a higher level of current harmonics and higher $K$ gains.

The second issue is related to the existence of harmonics in the grid voltage that could cause additional current harmonics in the passive part of the hybrid filter. This could be a problem when the passive part of a filter is tuned to the 5th or 7th harmonic because these harmonics are often viewable in distribution power systems. This could result in undesirable current flow between the grid and the hybrid active power filter, caused by series resonance for the selected harmonic. To prevent this, the control system of the converter, which measures the grid voltages, can ensure damping of undesirable currents by direct generation the higher harmonic of the grid voltage in power electronic part of HAPF—$v_{\mathrm{D}\alpha,\beta}$. These signals are filtered from the grid voltages $v_{\mathrm{G}}$ after the transformation to the synchronous rotating coordinate system and by using HPF filters.

## 4. Analysis of Time Delay Influence on Operation of Control System

One of the most important issues in the practical realization of active power filters and hybrid active power filters is the reduction of the time delay in the control system [64–66]. The time delay results from digital implementation of the control system and the usage of analogue filters in measurements. The CL1 control loop in HAPF is the closed control loop in which the time delay can cause operational instability. In Figure 5a the harmonic attenuation rates for HAPF with zero time delay and gains $K = 20$ and $K = 40$ are presented. As a reference, the harmonic attenuation rate for only the passive filter is shown.

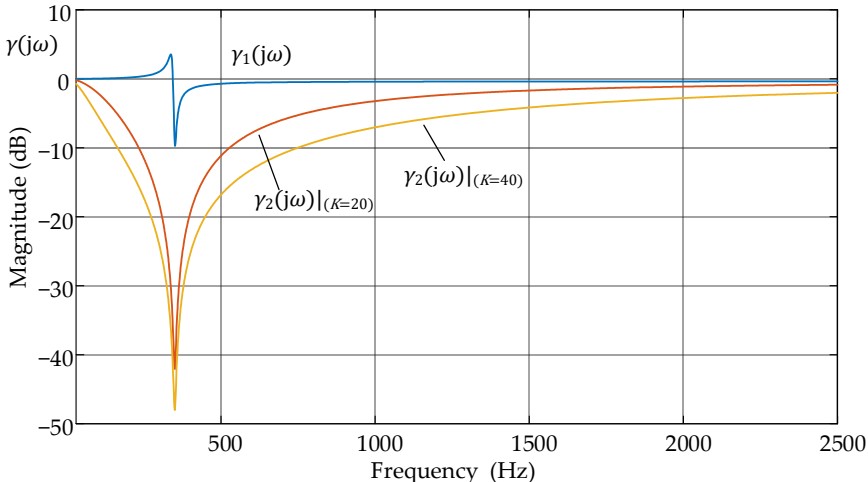

**(a)**

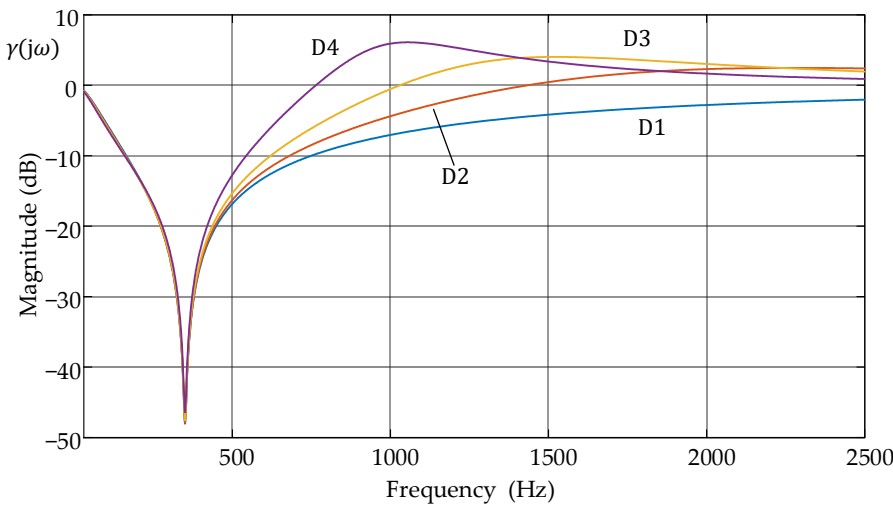

**(b)**

**Figure 5.** Harmonic attenuation rate of HAPF: (**a**) ideal operation condition (no time delay), passive filter PF ($\gamma_1$), HAPF with CL1 ($\gamma_2$), $K = 20$ and $K = 40$; (**b**) operation of HAPF with CL1 ($\gamma_2$) and $K = 40$ for different values of time delay, D1—$T_D = 0$ µs, D2—$T_D = 75$ µs, D3—$T_D = 150$ µs, D4—$T_D = 300$ µs.

The results presented were obtained in Matlab, using a simplified model based on transfer functions. It can be observed that in the frequency range of 0 to 2500 Hz, the harmonic attenuation rate is always lower than zero and higher values of gain $K$ ensure better operation of the HAPF.

Typically, in a microprocessor-based control system, the ADC (analog-to-digital converter) operation is synchronised with PWM (pulse width modulation) signals to prevent the influence of power electronic devices switching on the measurements. The time interval between the analog signals acquisition and setting new values of PWM is intended to perform the required computations (control loops, filtering, protection, communication). If the control algorithm is complex, the code needs to be optimized. It can be mainly assumed that the time for one cycle of computation is equal to the switching period. Additionally, the required PWM signals are generated in the next switching period, so the average time delay for the switching frequency $f_{sw}$ can be defined as:

$$T_D = \frac{3}{2f_{sw}} \tag{5}$$

When this time delay is taken into account, it can cause undesirable operation of the HAPF (Figure 5b). The time delay unexpectedly amplifies the harmonic attenuation rate of the HAPF in the analysed frequency range. This amplification of harmonics depends on the gain $K$ and the time delay $T_D$. The Figure 5b presents the harmonic attenuation rate of the HAPF when the CL1 control loop is applied for $K = 40$ with different time delays $T_D$. The harmonic attenuation rate is presented for D1: $T_D = 0$ μs, D2: $T_D = 75$ μs ($f_{sw} = 20$ kHz), D3: $T_D = 150$ μs ($f_{sw} = 10$ kHz) and D4: $T_D = 300$ μs ($f_{sw} = 5$ kHz). One can see that harmonic attenuation rates D3 and D4 can gain the value of current harmonics. The maximum value of harmonic attenuation rate increases with the time delay $T_D$ and the range of frequencies with positive attenuation rates is shifted towards the passive filter resonant frequency. For the switching frequency of 20 kHz, the HAPF can slightly amplify the harmonics of a frequency range above 1500 Hz.

The operation of the HAPF for time delay $T_D = 0$ μs and $T_D = 300$ μs is shown by current waveforms in Figure 6a,b, respectively. To present these results, the simplified model of the HAPF based on controlled voltage sources has been prepared in Matlab-Simulink. The blue waveform is the grid current $i_G$ and the red waveform is the load current $i_L$. When zero time delay is assumed, the HAPF operates accurately. When non-zero time delay is assumed, the amplification of the selected harmonic in the grid current can be observed. Generally, the gain $K$ in the CL1 control loop is limited by the time delay $T_D$ in a microprocessor-based control system.

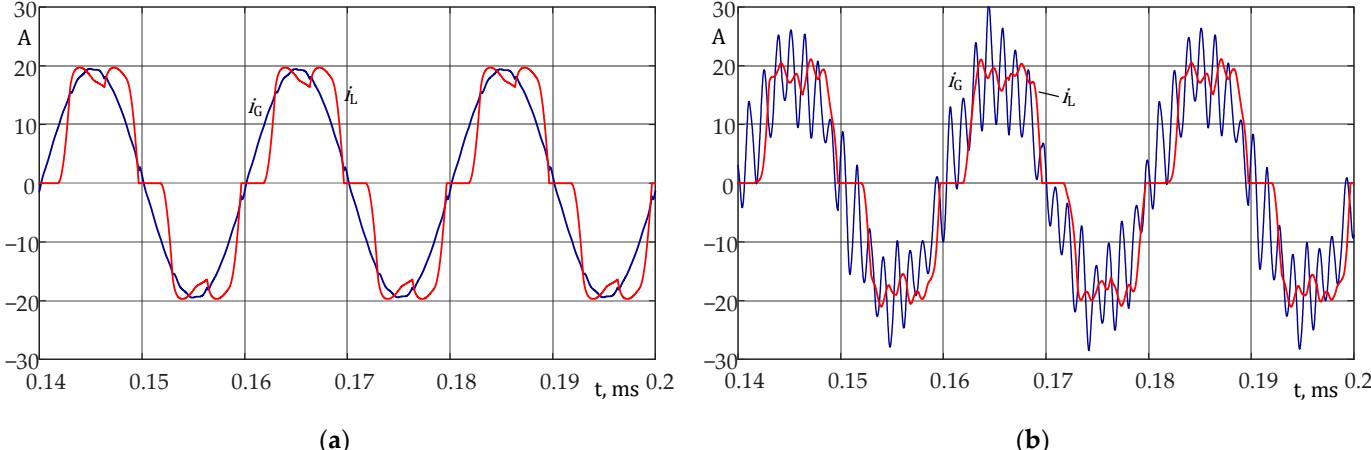

**(a)**          **(b)**

**Figure 6.** Simulations of operation of HAPF with CL1 control loop, $K = 40$: (**a**) with zero time delay $T_D = 0$ μs, (**b**) with the time delay $T_D = 300$ μs.

To ensure the small time delay, the execution of the control algorithm is divided into two parts, as shown in Figure 7. Typically, the control algorithm in the microprocessor is executed within one iteration of the interruption connected with PWM signal generation. During this iteration, the following sequence of operation is executed: (1) the acquisition of analog signals, (2) the execution of CL1 algorithm and CL2 algorithm, (3) the execution of additional computations (e.g., protection functions), (4) the setting of required PWM signals. This algorithm is referred to as Alg. 1 in Figure 7. For algorithm Alg.1 the time delay is given in (5). For the reduction of the time delay, another sequence has been proposed, which is referred as Alg. 2 and is shown in Figure 7. The start of the algorithm is at the maximum value of the counter in the PWM subcircuit. Firstly, the measurements of analog signals and CL1 algorithm are executed and then, based on the previous results of the CL2 control loop, the required PWM signals are computed. This procedure reduces the time delay for CL1 to $T_D = \frac{1}{f_{sw}}$ ($\frac{1}{2f_{sw}}$ for CL1 execution and $\frac{1}{2f_{sw}}$ for PMW signal generation) and allows to increase the gain $K$ in CL1. After that, the CL2 control algorithm is realized in the next interrupt procedure together with additional computations which are used in the computation of the forthcoming PWM signals. As previously mentioned, the time delay in

the CL2 control loop execution is not a critical issue. This is because the CL2 control loop constitutes a feedforward open loop control. Additionally, the time delay for CL2 can be compensated.

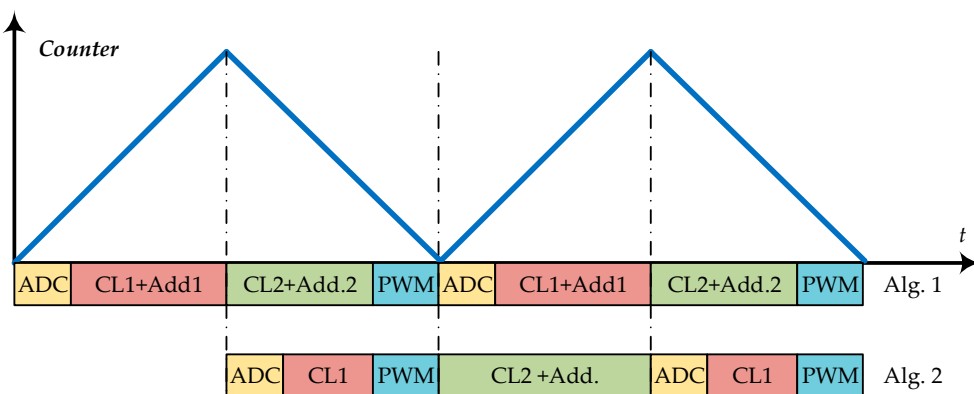

**Figure 7.** Execution of a control algorithm in microprocessor for reducing the time delay in CL1.

The CL2 control loop is based on the measurement of the load currents and its function is the generation of selected voltage harmonics which force current harmonics to flow through the passive filter (PF, Figure 4). By applying this approach, the selected grid current harmonics are reduced. Due to the fact that CL2 control loop is an open loop control, the time delay will not cause operation instability. However, the time delay will cause imperfect compensation.

The block diagram presenting the CL2 control for harmonic $h$ is depicted in Figure 8. Assuming that the time delay for CL2 realization is $\tau$, after the transformation to the rotating coordinate system and filtering with the use of the low pass filter LPF, the $d,q$ components will be phase shifted with the shift angle proportional to the time delay and the harmonic order $h$. For example, for the time delay $\tau = 100$ μs, phase shift angle of the 5th harmonic will be equal to 9 degrees, but for the 17th harmonic the phase shift angle will be equal to 30 degrees, which shows that it could cause improper operation of CL2 for higher order harmonics. To avoid such a phase shift, the additional compensation block CO has been added into the CL2 control loop. This block represents the operation of rotation for the phase shift but with an opposite direction. It has to be noted that CO block takes into account the sequence of the harmonic in the three-phase system (positive or negative sequence). It can be seen that the operation of rotation CO is similar to the transformation from the stationary to the rotating coordinate system, but with constant coefficients for $h$ harmonic. Assume that impedance matrices $Z_{Fh}$ are defined as:

$$Z_{Fh} = \begin{bmatrix} \mathrm{Re}\{Z_{PF}(j\omega_h)\} & -\mathrm{Im}\{Z_{PF}(j\omega_h)\} \\ \mathrm{Im}\{Z_{PF}(j\omega_h)\} & \mathrm{Re}\{Z_{PF}(j\omega_h)\} \end{bmatrix}, \tag{6}$$

where Re is a real part of impedance of passive filter for $h$ harmonic and Im is an imaginary part of impedance of passive filter for $h$ harmonic, the CO can be described for a particular harmonic as:

$$CO_h = \begin{bmatrix} \cos(h\omega\tau) & -\sin(h\omega\tau) \\ \sin(h\omega\tau) & \cos(h\omega\tau) \end{bmatrix}, \tag{7}$$

where $h$ is positive for positive sequence harmonic and negative for negative sequence harmonic. Based on both (6) and (7), the operation of compensation and multiplication by the impedance matrix, can be replaced by a single matrix operation with the modified impedance matrix $Z_{Fh}{}^K$. The elements of the modified matrix $Z_{Fh}{}^K$ can be computed offline and the execution of CL2 with compensation CO will not take more time than the execution of the CL2 control loop without the compensation.

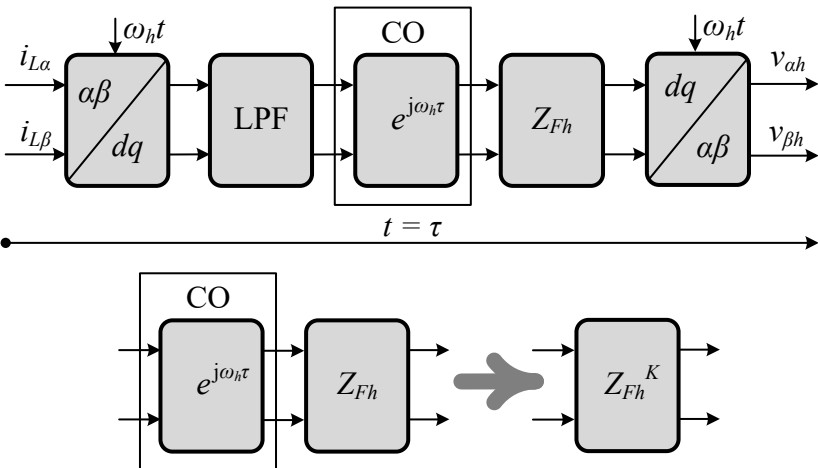

**Figure 8.** The modification of the CL2 control loop.

The effect of the compensation is verified in a simulation model where only CL2 control loop is used (without the CL1). The time delay $\tau$ is 100 µs. The simulation results are presented in Table 1 and in Figure 9. Table 1 presents *THD* of the currents and levels of selected harmonics of load current and the grid currents. The table presents results with and without the compensation of time delay.

**Table 1.** Operation comparison for hybrid active power filters (HAPF) with CL2 control loop without and with time delay compensation.

| Harmonic | Load Current | Grid Current, CL2 without Delay Compensation | Grid Current, CL2 with Delay Compensation |
|:---:|:---:|:---:|:---:|
| 1 | 15.3 A/100% | 14.8 A/100% | 14.9 A/100% |
| 5 | 10.0 A/65.7% | 2.3 A/15.5% | 0.24 A/1.6% |
| 7 | 6.3 A/41.3% | 0.60 A/4.1% | 0.60 A/3.7% |
| 11 | 1.3 A/8.3% | 0.36 A/2.4% | 0.04 A/0.3% |
| 13 | 1.2 A/7.6% | 0.40 A/2.7% | 0.04 A/0.3% |
| 17 | 0.76 A/5.0% | 0.36 A/2.4% | 0.03 A/0.4% |
| *THD* | 78.8% | 17.1% | 5.3% |

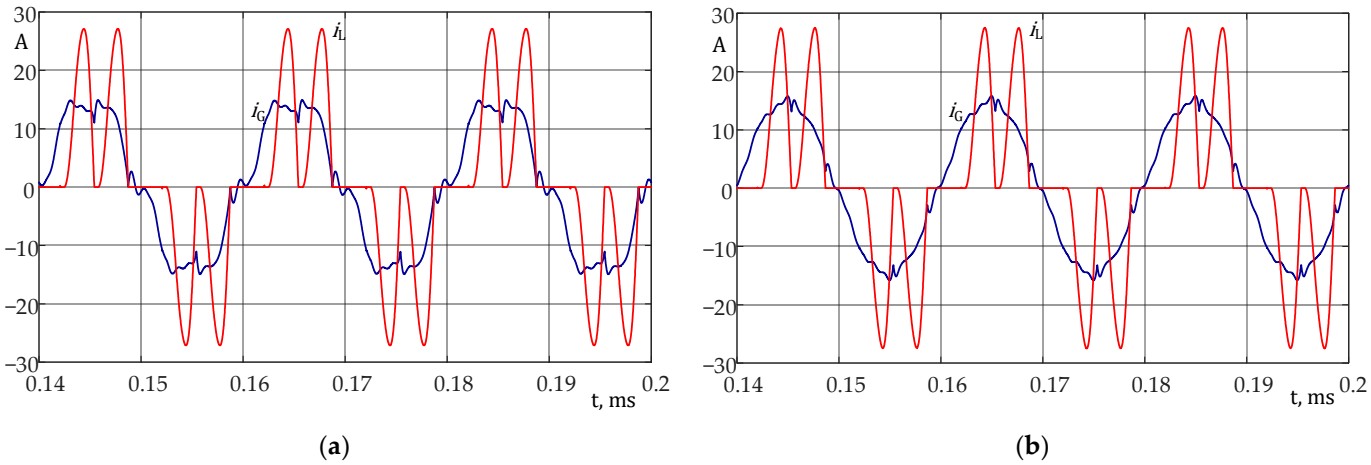

(**a**)　　　　　　　　　　　　　　　　　(**b**)

**Figure 9.** Operation of HAPF with CL2 control loop: (**a**) without and (**b**) with the time delay compensation.

It can be seen that for the operation of CL2 with and without the time delay compensation for the 7th harmonic (which is the harmonic for PF tuning), the same reduction level is achieved. For other harmonics, reduction is lower than in the case of operation without time delay compensation. It is caused by the phase shift in the transformation to *d,q* coordinate system. The *THD* of the grid current for operation without time delay compensation is 17.1%, while with the time delay compensation the *THD* can be reduced to 5.3%. It has to be mentioned that when the HAPF operates with both the CL1 control loop (*K* = 40) and CL2 with the time delay compensation, the grid current *THD* is further reduced to 3.7%. Figure 9 presents the load currents and the grid current for an operation of HAPF with CL2 control loop for conditions presented in Table 1. It can be seen that the time delay compensation ensures better results.

## 5. Parameter Selection for the Control Algorithm

The important aspect during the design stage of the HAPF is the parameter selection of the control algorithm. As presented in this section, the control algorithm parameters have a significant impact on the effectiveness and stability of the HAPF operation.

### 5.1. Parameter Selection for dc Link Voltage Controller

In the analyzed control system, the dc link voltage controller plays only a secondary role. Its main task of this controller is to keep the dc link voltage at the reference value $V_{\mathrm{DCREQ}}$. However, too fast a dynamic response of the dc voltage controller may result in distortion of the generated currents. In the control algorithm, the classical PI controller has been proposed (Figure 4). By taking into account the HAPF with its control algorithm it is possible to derive the simple dynamic model with the dc link voltage controller (Figure 10).

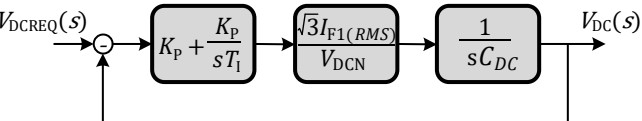

**Figure 10.** Dynamic model of HAPF dc link voltage regulation.

The transfer function of the presented model can be expressed as follows:

$$\frac{V_{\mathrm{DC}}(s)}{V_{\mathrm{DCREF}}(s)} = \frac{T_{\mathrm{I}}s + 1}{G_{\mathrm{DC}}s^2 + T_{\mathrm{I}}s + 1},\tag{8}$$

$$G_{\mathrm{DC}} = \frac{C_{\mathrm{DC}}T_{\mathrm{I}}V_{\mathrm{DCN}}}{\sqrt{3}K_{\mathrm{P}}I_{\mathrm{F1(RMS)}}},\tag{9}$$

where: $K_{\mathrm{P}}$—proportional gain, $T_{\mathrm{I}}$—integrating time of PI controller, $I_{\mathrm{F1(RMS)}}$— root mean square value (RMS) of the fundamental component of the passive filter current, $C_{\mathrm{DC}}$—dc link capacitance, $V_{\mathrm{DCN}}$—dc link rated voltage.

From (8) and (9) one can derive the damping factor $\xi$ and the condition, for which the step response of the PI controller is critically damped:

$$\xi = \frac{T_{\mathrm{I}}}{2\sqrt{G_{\mathrm{DC}}}} = \frac{1}{2}\sqrt{\frac{\sqrt{3}K_{\mathrm{P}}T_{\mathrm{I}}I_{\mathrm{F1(RMS)}}}{C_{\mathrm{DC}}V_{\mathrm{DCN}}}} \geq 1.\tag{10}$$

The step responses of the simple dynamic model with the dc link voltage controller are shown in Figure 11 for different parameters of the PI controller but for the damping factor $\xi$ = 1. It is assumed that the integrating time $T_{\mathrm{I}}$ is at the level of the fundamental period i.e., from 20 ms to 50 ms. The integrating time of $T_{\mathrm{I}}$ = 40 ms is chosen for the control in the experimental prototype. For such $T_{\mathrm{I}}$ the controller proportional gain is equal to $K_{\mathrm{P}}$ = 3.5. During the experiments due to the observed noise in measured signals the value of the proportional gain $K_{\mathrm{P}}$ has been reduced to $K_{\mathrm{P}}$ = 1.

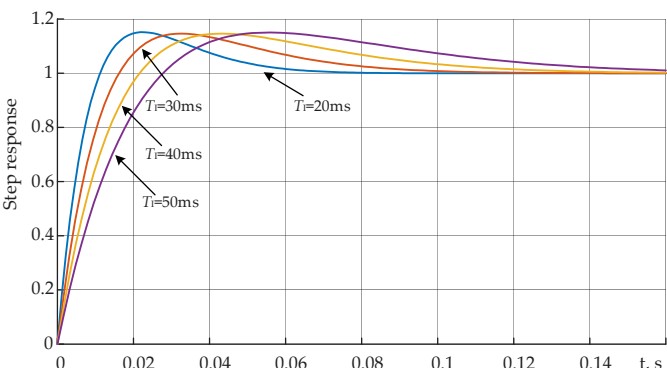

**Figure 11.** Step responses of the dc link voltage for different integrating times $T_\mathrm{I}$ and the damping factor $\xi = 1$.

### 5.2. Parameter Selection for Signal Filters in Current Control Loops

In both current control loops the signal low-pass filter (LPF) and the high-pass filter (HPF) are applied. Their role is to attenuate time-varying or dc components of the signal dq components. These signal filters are mainly responsible for the dynamic response to the load current changes. It is important to select filters with a fast step response, therefore the filter order should not be too high. Both signal filters have to have appropriate attenuation rate in the chosen bandwidth.

Both low-pass and high-pass filters are selected as 2nd order Butterworth filters with the following transfer functions:

$$K_\mathrm{LPF}(\mathrm{j}\omega) = \frac{\omega_\mathrm{c}^2}{(\mathrm{j}\omega)^2 + \frac{\omega_\mathrm{c}}{Q}(\mathrm{j}\omega) + \omega_\mathrm{c}^2} \ , \tag{11}$$

$$K_\mathrm{HPF}(\mathrm{j}\omega) = \frac{(\mathrm{j}\omega)^2}{(\mathrm{j}\omega)^2 + \frac{\omega_\mathrm{c}}{Q}(\mathrm{j}\omega) + \omega_\mathrm{c}^2} \ , \tag{12}$$

where $Q$ factor is equal $\sqrt{2}/2$ and $\omega_\mathrm{c}$ is the cut-off angular frequency, $\omega_\mathrm{c} = 2\pi f_\mathrm{c}$.

The selection of the cut-off frequency $f_\mathrm{c}$ is done as a trade-off between a relatively high attenuation rate of unwanted harmonics and a fast step response of both filters. Therefore, the selected cut-off frequency is $f_\mathrm{c} = 25$ Hz. Figure 12 shows the step responses together with the Bode magnitude plots of the low-pass filter for different values of the cut-off frequency $f_\mathrm{c}$. It is clear that for the higher cut-off frequency its step response is faster but the LPF gain obtained from the frequency response plot (Figure 12b) is higher in the required bandwidth. The analysis for the high-pass filter is similar.

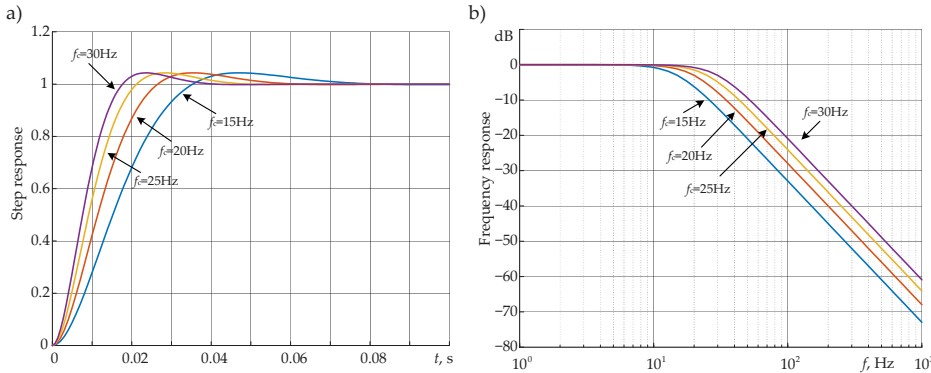

**Figure 12.** Step (**a**) and frequency (**b**) responses of low pass filter for different cut-off frequencies $f_\text{c}$.

### 5.3. Selection of the Gain K for the Hybrid Active Power Filter

As mentioned earlier, the gain *K*, which is applied in the current control loop CL1, has an impact on the attenuation rate for current harmonic reduction (as in Figure 5a). The gain *K* also influences on the stability of the control loop. In Figure 13 one can see the simplified dynamic model of control loop CL1, which allows stability analysis to be performed.

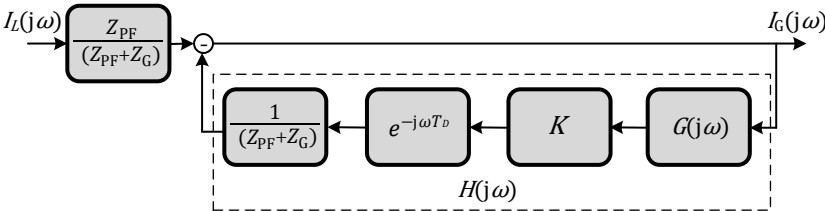

**Figure 13.** Frequency model used in stability analysis.

The transfer function of the feedback loop, shown in Figure 13, is given as:

$$H(\text{j}\omega) = \frac{KG(\text{j}\omega)\text{e}^{-\text{j}\omega T_\text{D}}}{Z_\text{PF}(\text{j}\omega) + Z_\text{G}(\text{j}\omega)} \, , \tag{13}$$

where: $T_\text{D}$—the time delay of the control loop CL1, which is equal to 50 µs, $G(\text{j}\omega)$, a transfer function similar to the harmonic attenuation rate given by (3) but taking into account the transfer function $G_\text{HPF-dq}(\text{j}\omega)$ which represents a part of the CL1 loop responsible for detecting the higher order harmonics.

$$G(\text{j}\omega) = \frac{Z_\text{PF}(\text{j}\omega)}{Z_\text{PF}(\text{j}\omega) + Z_\text{G}(\text{j}\omega) + G_\text{HPF-dq}(\text{j}\omega)K} \, , \tag{14}$$

The transfer function $G_\text{HPF-dq}(\text{j}\omega)$ represents both the high-pass filter (HPF) and dq transformation [62]. The transfer function $G_\text{HPF-dq}(\text{j}\omega)$ depends on the sequence of symmetrical components. For the sake of simplification of the analysis only the transfer function for positive sequence components is shown as:

$$G_\text{HPF-dq}(\text{j}\omega) = \frac{(\text{j}\omega - \text{j}\omega_1)^2}{(\text{j}\omega - \text{j}\omega_1)^2 + \frac{\omega_\text{c}}{Q}(\text{j}\omega - \text{j}\omega_1) + {\omega_\text{c}}^2} \, , \tag{15}$$

where $Q$ and $\omega_\text{c}$ are the parameters of the HPF mentioned in the Section 5.2.

Based on the frequency model of the control loop CL1 (Figure 13) it is possible to perform the stability analysis by using the Nyquist stability criterion. The stability analysis is performed for different values of the gain *K* and examples of the results are shown in Figure 14. As is seen in Figure 14b for the gain *K* = 70 the analyzed model is unstable, therefore in this paper lower values of the gain *K* are chosen. For the gain *K* = 35 the

system is stable (Figure 14a) and guarantees satisfactory effectiveness of the fifth harmonic reduction (Figure 5a).

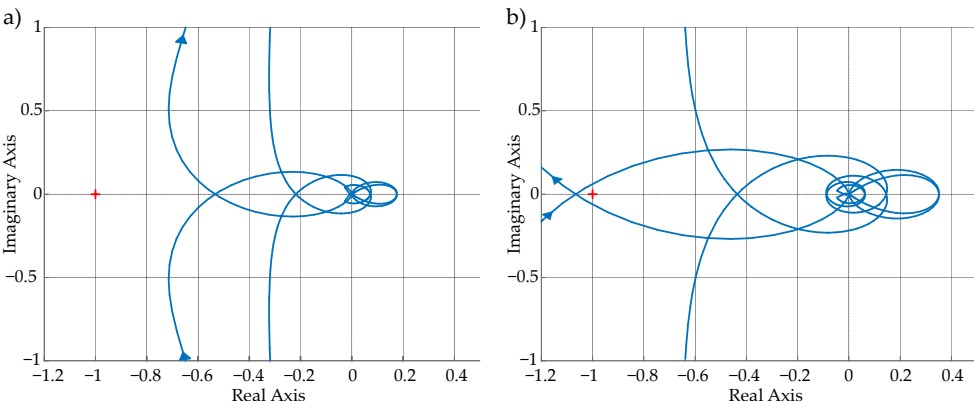

**Figure 14.** Nyquist diagram for different *K* (**a**) *K* = 35, (**b**) *K* = 70.

## 6. Experimental Results

The experimental tests have been carried out on the prototype system of the hybrid active power filter presented in Figure 15. The prototype has been placed inside the explosion-proof housing for the mining applications. Figure 15 presents a 3D model of the HAPF and its prototype photograph.

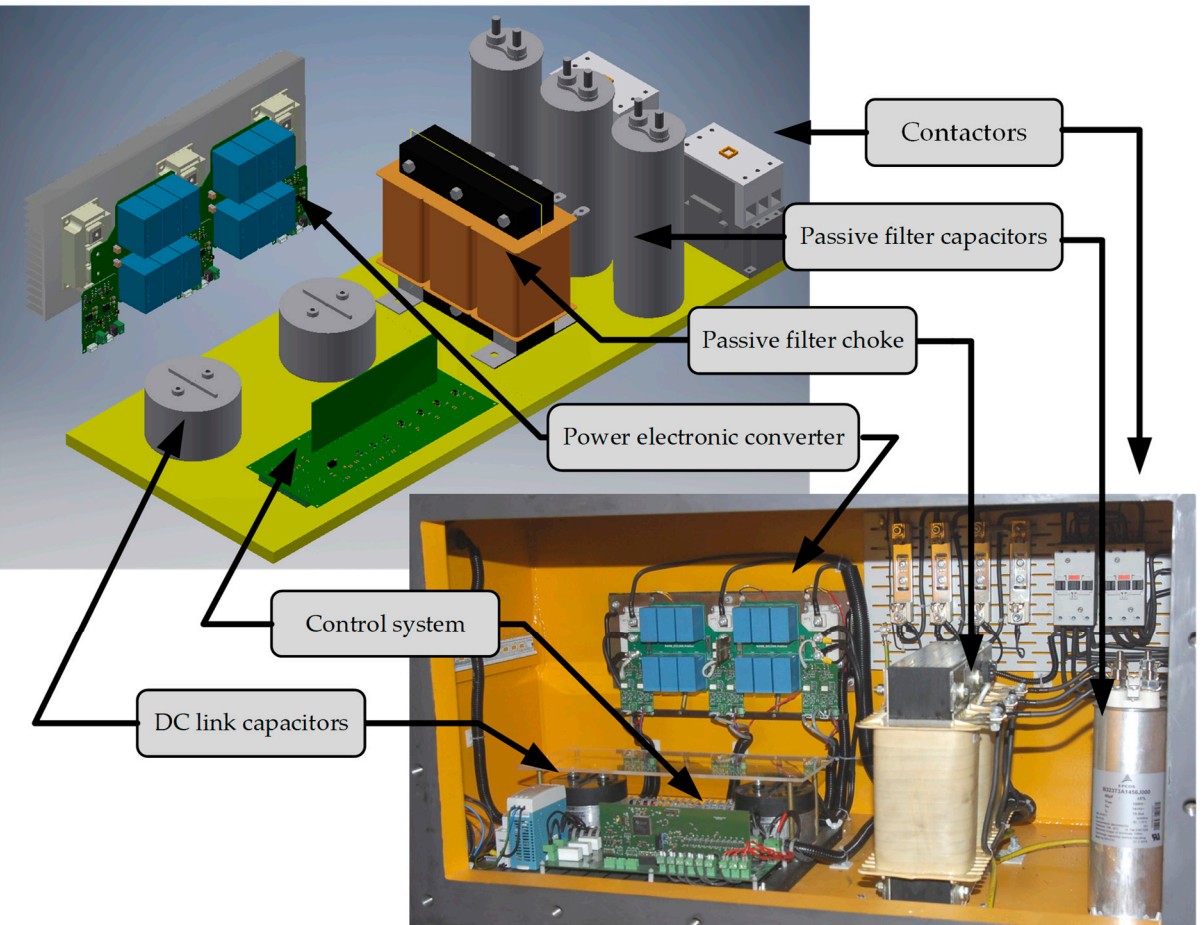

**Figure 15.** Prototype of hybrid active power filter inside the explosion-proof housing.

The prototype consists of the passive part (filter chokes and capacitors) and the power electronic converter with its control system. The power electronic converter is placed on the left side. The converter consists of three SiC CREE 1200 V half-bridge modules CAS300M12BM2 with nominal $R_{DS(on)}$ = 4.2 m$\Omega$. These modules are used for ensuring low power losses due to the limitation on power dissipation of the explosion proof housing.

The dc circuit consists of specially matched PCBs with auxiliary capacitors and two main capacitors. To take full advantage of their capabilities, a dc link with minimized inductance has been designed. It uses low inductance polypropylene capacitors mounted on PCBs matched to the leads of transistor modules. As the main dc link capacitance, two 600 µF capacitors are used.

Subsequent PCBs serve as transistor drivers and are also attached directly to the SiC power modules as an interface to the control system.

The microprocessor-based controller is placed on the separate PCB board. This controller utilizes a 32-bit floating point microcontroller TMS320F28335. The control system ensures the measurement of all required analog signals, generation of PWM signals, protection and MODBUS RTU communication. The control system operates with 20 kHz switching frequency which was set at this value as a trade-off between the fast operation of the HAPF and its low switching losses.

The passive filter consisting of a choke and three AC capacitors is placed on the right side. The passive filter is tuned to 7th order harmonic, which ensures the reduction of the passive filter size and its weight. On the far right side, there are contactors (the main and start-up contactor) and protection devices (fuses). Parameters of the prototype of HAPF are given in Table 2.

**Table 2.** Rated parameters of the prototype of the hybrid active power filter HAPF.

| Parameter | Description | Value |
|---|---|---|
| $V_N$ | Rated line-to-line voltage RMS value | 1000 V |
| $I_N$ | Rated phase current RMS value | 30 A |
| $V_{DCN}$ | dc link rated voltage | 400 V |
| $f_{sw}$ | Switching frequency | 20 kHz |
| $L_F$ | Passive filter inductance | 4.6 mH |
| $C_F$ | Passive filter capacitance | 45 µF |
| $C_{DC}$ | dc link capacitance | 1.2 mF |
| $K$ | Gain factor | 25–35 |
| $K_P$ | dc voltage controller proportional gain | 1 |
| $T_I$ | dc voltage controller integration time | 40 ms |
| $f_c$ | LPF and HPF cutoff frequency | 25 Hz |

*6.1. Verification of Appropriateness of the Control Algorithm Modifications*

As previously mentioned, in case of distortion of the grid voltage, the passive part of the hybrid active power filter can increase the grid current harmonics. Figure 16a shows the results of operation of the passive filter with the power electronic converter generating zero voltage (which corresponds to the short circuit of converter terminals). The grid voltage *THD* is 2.8%. This voltage is distorted mainly by the 5th and 7th harmonics. One can see that the passive filter generates a grid current containing mainly the 7th harmonic with *THD* = 43.4%. By adding additional signals $v_{D\alpha,\beta}$ (which correspond to measured grid voltage) to the control system, the *THD* of the grid currents is reduced to 7.1%. Applying the CL1 control loop with the gain $K$ = 25 further reduces the *THD* to 3.5%. The results of this compensation are presented in Figure 16b. From the waveforms of the current and voltage in Figure 16b one can see that only reactive power exists. This is caused by capacitors of the passive filter PF, and reveals the inherent feature of the hybrid active power filter. The reactive power cannot be fully compensated due to the limited level of converter dc link voltage $V_{dc}$. Such a reactive power can be used for the load reactive power compensation. The effect of the time delay compensation in the CL2 control loop is shown in Figure 17. The waveforms are presented for the HAPF operation with the

non-linear load with the current THD equal to 27.0%. In Figure 17a, the grid and load currents together with its harmonic spectra are shown for the HAPF operation without the time delay compensation. The harmonic reduction is presented for the HAPF operation with both CL1 and CL2 control loops. The gain $K = 35$ is set for control loop CL1 and the CL2 control loop operates with the elimination of 5th, 11th, 13th, 17th harmonics. It can be seen that for the 5th and 11th harmonics, the CL2 control loop operates quite well, but for 13th and 19th harmonics the results are not satisfactory. Figure 17b presents the waveforms for the operation of HAPF with the same gain $K$ in CL1 control loop and the time delay compensation in CL2 control loop. Thanks to the time delay, compensation harmonics in the grid current are at lower levels (this is particularly true for the 13th and 17th harmonics). The shape of the grid current is better and its THD is reduced to 4.6% instead of 7.7 % for operation without the time delay compensation from Figure 17a.

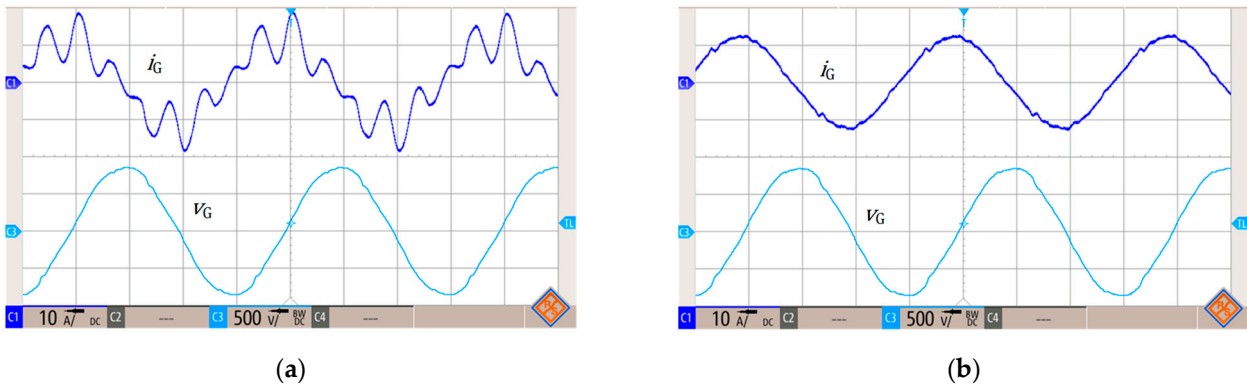

(**a**)      (**b**)

**Figure 16.** Compensation of the grid voltage distortions in hybrid filter: (**a**) operation without compensation with $K = 0$; (**b**) operation with voltage distortion compensation and $K = 25$.

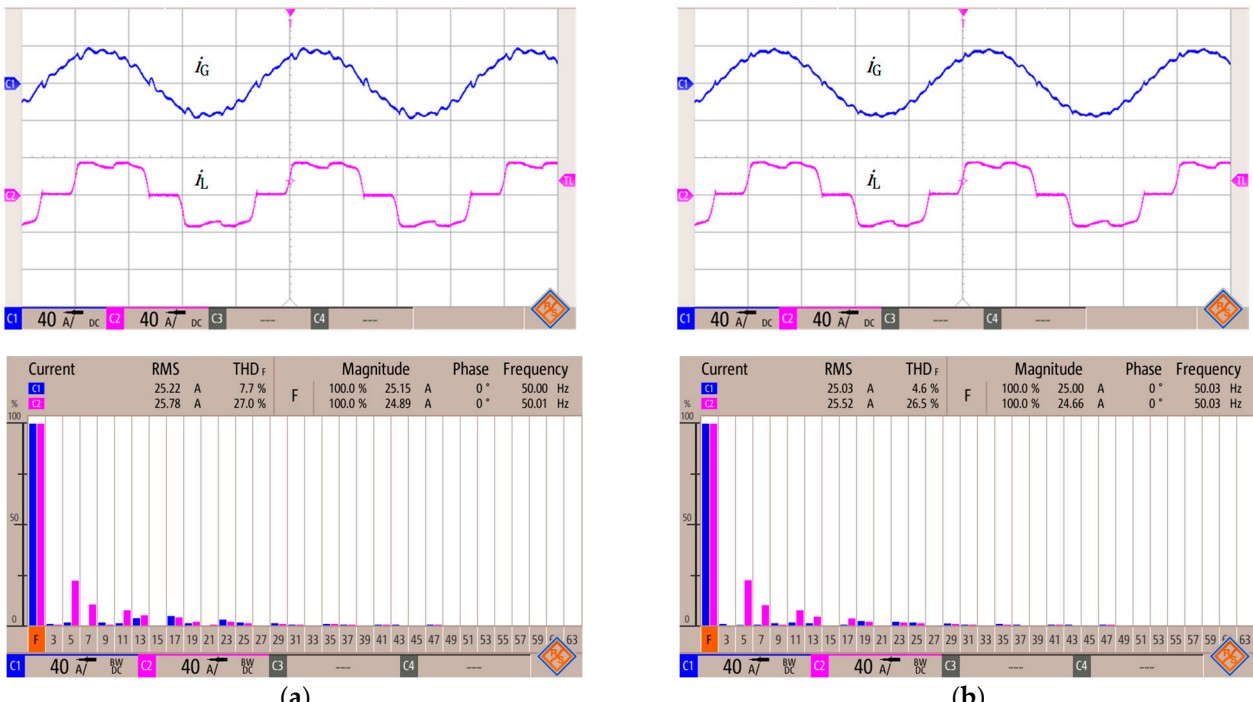

(**a**)      (**b**)

**Figure 17.** The effect of the time delay compensation on current waveforms and their spectra: (**a**) operation of HAPF without the time delay compensation; (**b**) operation of HAPF with the time delay compensation.

### 6.2. Harmonics Reduction Performance for Different Loads

The operation of HAPF with different loads is presented in Figures 18–20. Figures 18a, 19a and 20a present waveforms of the grid current $i_G$, the load current $i_L$ and filtered (by using a low pass filtering of PWM modulation) power electronic converter voltage $v_{PC}$. Figures 18b, 19b and 20b present harmonic spectra of the grid and the load currents. Figure 18 depicts the operation of the HAPF with the load Ld1 characterized by high current distortions (*THD* = 85.5%, *P* = 59 kW) with a high content of 5th, 7th and 11th harmonics. The load Ld1 represents typical AC drives or diode rectifiers with parallel RC connection in the dc link (relatively high dc link capacitance with a small input inductance). The HAPF operates with both control loops as presented in Figure 17b. One can see that the HAPF is able to reduce the *THD* from 85.5% to 3.8%. In this case, the voltage generated in the power electronic converter includes mainly harmonics with orders $h < 11$. The harmonic amplitudes are listed in Table 3. The RMS value of the grid current $i_G$ is reduced in comparison to the load current $i_L$ but the grid current consists of both active and reactive components.

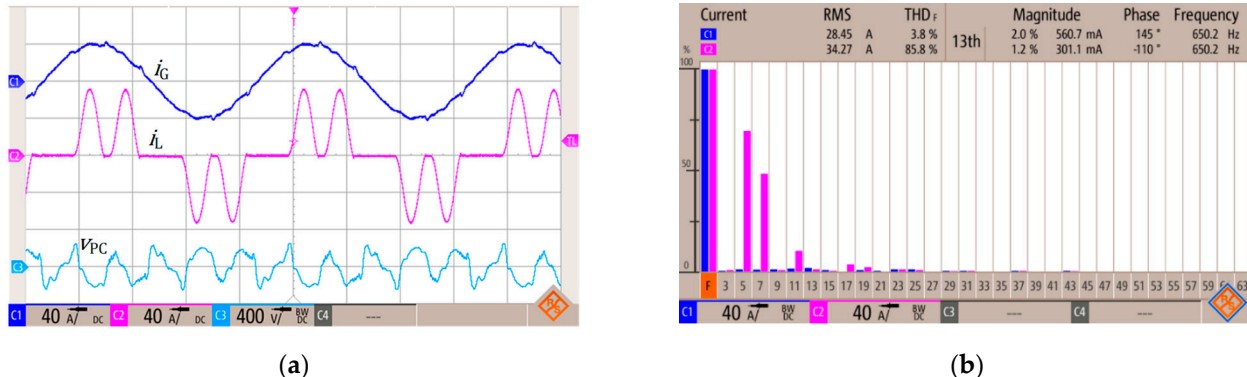

(**a**)  (**b**)

**Figure 18.** Operation of HAPF with the load Ld1: (**a**) the grid current, the load current and the power converter voltage waveforms; (**b**) harmonic spectrum of the grid and the load currents.

**Table 3.** Comparison of the load currents and the grid currents for HAPF with the load Ld1.

| Harmonic | Load Current | Grid Current |
|:---:|:---:|:---:|
| 1 | 26.0 A/100% | 28.4 A/100% |
| 5 | 18.2 A/70.0% | 0.36 A/1.3% |
| 7 | 12.7 A/48.8% | 0.32 A/1.1% |
| 11 | 2.81 A/10.8% | 0.47 A/1.7% |
| 13 | 0.30 A/1.2% | 0.56 A/2.0% |
| 17 | 0.97 A/3.7% | 0.03 A/0.1% |
| 19 | 0.58 A/2.2% | 0.3 A/1.0% |
| *THD* | 85.8% | 3.8% |

The operation of HAPF with a diode rectifier with R load in the dc link with line reactors is presented in Figure 19. The load Ld2 (*P* = 44 kW) is characterized by lower level of *THD* (26.6%), the levels of harmonics are lower in comparison to the Ld1 load, but the spectrum of harmonics is significantly wider. For such conditions, the HAPF reduces the *THD* of the grid current to 4.0%. The values of selected current harmonics are listed in Table 4. For the load Ld2, the voltage generated by the power electronic converter includes more higher order harmonics than for the load Ld1.

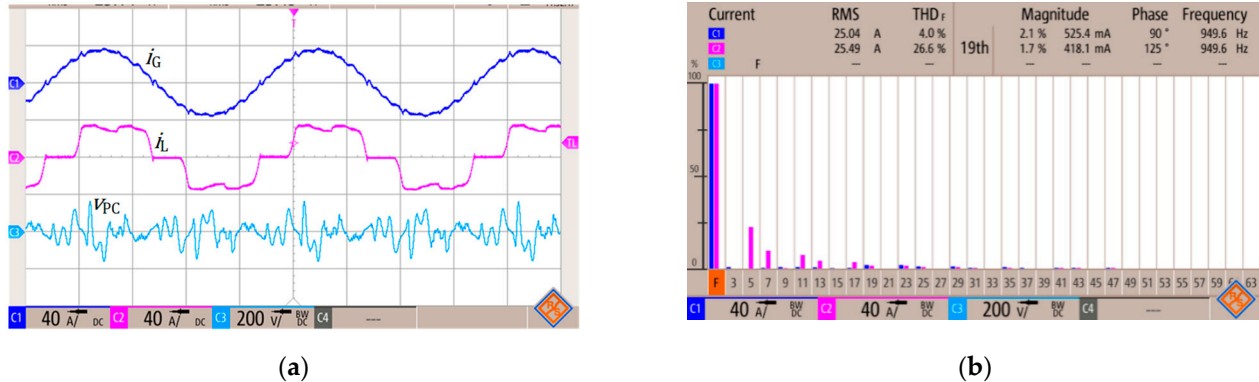

(**a**)  (**b**)

**Figure 19.** Operation of HAPF with the load Ld2: (**a**) the grid current, the load current and the power converter voltage waveforms; (**b**) harmonic spectrum of the grid and the load currents.

**Table 4.** Comparison of the load currents and the grid currents for HAPF with the load Ld2.

| Harmonic | Load Current | Grid Current |
|----------|-------------|-------------|
| 1 | 24.6 A/100% | 25.0 A/100% |
| 5 | 5.58 A/22.7% | 0.08 A/0.3% |
| 7 | 2.47 A/10.0% | 0.11 A/0.4% |
| 11 | 1.88 A/7.6% | 0.29 A/1.2% |
| 13 | 1.19 A/4.5% | 0.19 A/0.8% |
| 17 | 0.9 A/3.7% | 0.17 A/0.7% |
| 19 | 0.42 A/1.7% | 0.52 A/2.1% |
| *THD* | 26.0% | 4.0% |

The operation of the HAPF with the diode rectifier with RC load having smaller capacitance than in the case of the load Ld1 and with the line reactors similar to case Ld2 is depicted in Figure 20. The load Ld3 (*P* = 46 kW) currents are characterized by *THD* = 39.4% and for such a type of the load the HAPF reduces the grid current *THD* to 4.2%. The values of selected harmonics are presented in Table 5.

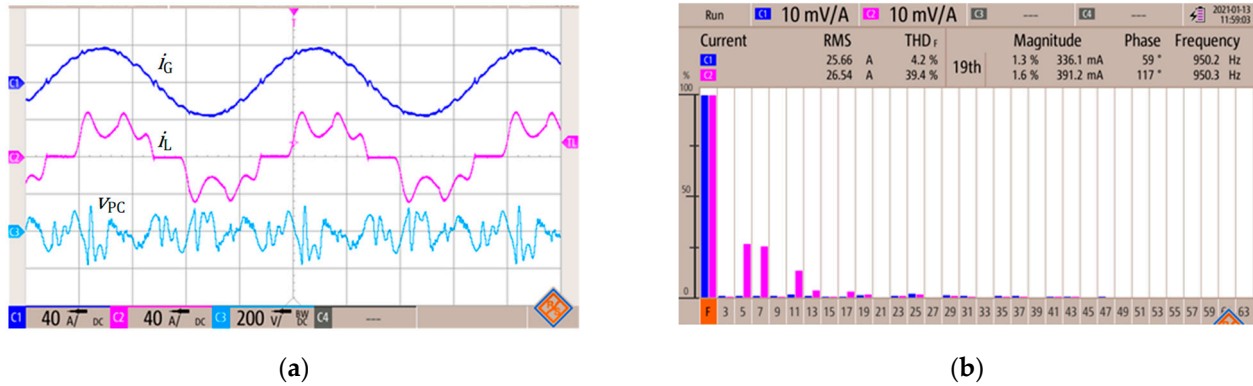

(**a**)  (**b**)

**Figure 20.** Operation of HAPF with the load Ld3: (**a**) the grid current, the load current and the power converter voltage waveforms; (**b**) harmonic spectrum of the grid and the load currents.

**Table 5.** Comparison of the load currents and the grid currents for HAPF with the load Ld3.

| Harmonic | Load Current | Grid Current |
|:---:|:---:|:---:|
| 1 | 24.7 A/100% | 25.6 A/100% |
| 5 | 6.57 A/26.6% | 0.25 A/1.0% |
| 7 | 6.29 A/23.5% | 0.21 A/0.9% |
| 11 | 3.35 A/13.5% | 0.38 A/1.5% |
| 13 | 0.9 A/3.6% | 0.21 A/0.9% |
| 17 | 0.82 A/3.3% | 0.1 A/0.4% |
| 19 | 0.41 A/1.7% | 0.34 A/1.3% |
| *THD* | 39.4% | 4.2% |

It can be seen that the HAPF successfully reduces all harmonics included in the control system. The presented results verified the correct operation of the HAPF for different loads and the effectiveness of control improvements described in Section 4.

*6.3. Transient Tests*

Figure 21 shows the waveforms during transient at the load. It is clearly visible that after less than 20 ms, the HAPF correctly reduces harmonics. This directly results from the cut-off frequency of the signal filters (LPF and HPF). A longer regulation time of about 80 ms is observed at the dc link voltage. As expected, the dc link voltage variations generates a fundamental harmonic frequency component in the output voltage ($v_{PC}$). The changes in $v_{DC}$ in this case are not more than 5% of the dc link rated voltage.

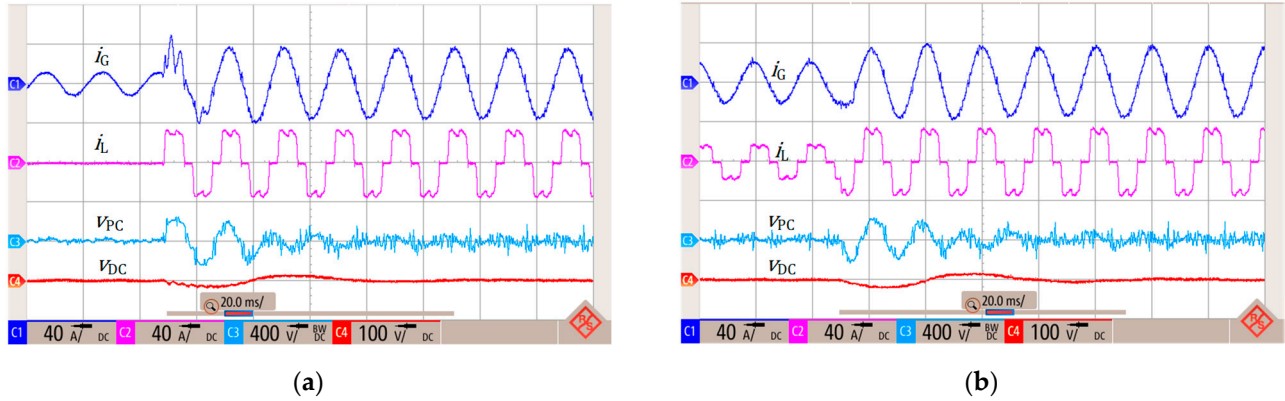

|  (a)  |  (b)  |

**Figure 21.** Example waveforms for the HAPF dynamic states (**a**) start the load Ld1 (**b**) power change of load Ld1.

*6.4. Power Losses*

As previously mentioned, the converter power losses play a crucial role, particularly in applications where utilization of the cooling system is limited. Therefore, it is required to identify the contribution of the HAPF components on total power losses. This contribution has been inspected by measuring power losses generated inside the converter and inside the passive filter. During the power loss measurements, HAPF has been operated with a load Ld2 with varied resistance in the dc-link. The results of power losses, which are measured by using the precise power analyzer WT5000E, are presented in Figure 22.

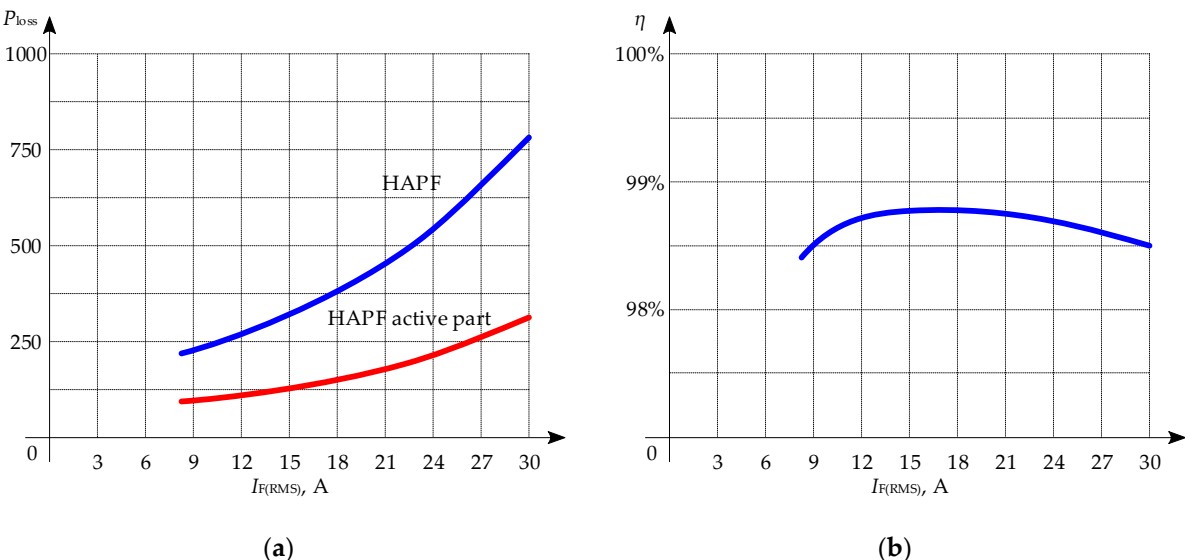

**Figure 22.** Measurement results of HAPF power characteristics given as a function of the RMS value of the filter current: (**a**) power losses of the active and passive part of HAPF; (**b**) the efficiency of the HAPF.

From Figure 22 one can see that power losses generated inside of the active part of the HAPF system, which is the converter part, are at least two times smaller compared to the passive part power losses. For RMS value of the current equal to 30 A, power losses generated in the converter are equal to 308 W, but in the passive part these losses are equal to 773 W. The efficiency of the HAPF is calculated from the following formula:

$$\eta = 1 - \frac{P_{\text{loss}}}{S},\qquad(16)$$

where $P_{\text{loss}}$ are power losses generated inside of the power electronic converter or in the whole hybrid active power filter and $S$ is the apparent power given as:

$$S = \sqrt{3}V_{(\text{RMS})}I_{\text{F(RMS)}},\qquad(17)$$

where $V_{(\text{RMS})}$ and $I_{\text{F(RMS)}}$ are line-to-line voltage and current RMS values of HAPF, respectively.

## 7. Conclusions

The hybrid active power filter for mining applications has been proposed. This implementation utilizes an original control method, which allows harmonic currents in the grid to be reduced. This method is based on the time delay compensation and voltage distortion compensation. Moreover, a change of the sequence in microprocessor computations has been proposed. Theoretical considerations of the proposed hybrid active power filter have been analyzed and experimentally verified. In all analyzed cases the proposed hybrid active power filter reduces harmonics to acceptable levels. Moreover, the proposed structure of the hybrid active power filter based on SiC transistors allows small power losses to be achieved. This enables the HAPF to be incorporated in an explosion-proof housing and thus used in the mining industry.

**Author Contributions:** Conceptualization, D.B. and J.M.; methodology, D.B., J.M., M.Z.; software, D.B. and J.M.; validation, D.B., J.M., M.Z., T.A., G.J. and M.J.; formal analysis, D.B. and J.M.; writing—original draft preparation, D.B., J.M., M.Z., T.A. and G.J.; writing—review and editing, D.B., J.M., M.Z., T.A., G.J. and M.J.; supervision, D.B. and J.M. All authors have read and agreed to the published version of the manuscript.

**Funding:** This research was partly financed by the EU funding for 2014–2020 within Smart Growth Operational Programme.

**Institutional Review Board Statement:** Not applicable.

**Informed Consent Statement:** Not applicable.

**Data Availability Statement:** Data sharing not applicable.

**Conflicts of Interest:** The authors declare no conflict of interest.

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
