# Peer review of "Control Strategy of 1 kV Hybrid Active Power Filter for Mining Applications"

_energies, doi:10.3390/en14164994_

Round 1
Reviewer 1 Report
This paper deals with a very interesting and useful proposal of a hybrid active power filter for real-world industry application (i.e. mining). The paper is well-written (but needs English editings) and supported by experimental validations.
The following are some lacks to be addressed:
- The authors statement that their control strategy is innovative is not supported by any comparison versus well-known active power filter methods. This should be well highlighted.
- If the proposal has been tested in real mining conditions, this should be mentioned and illustrated by photos (experiments context).
Author Response
The authors would like to thank the Reviewer for the effort in reviewing the paper. Please be advised that the English language proofreading of the paper has been done.
Comment 1: The authors statement that their control strategy is innovative is not supported by any comparison versus well-known active power filter methods. This should be well highlighted.
Answer: The aim of the article was not to compare the proposed control strategy to other methods, therefore there is no such direct comparison in the text. Nevertheless, the proposed strategy is based on the well-known and cited method with feedback and feedforward control. Therefore, the paper was limited to show how the implemented improvements affect the effectiveness of HAPF. This is given in section 6.1. (the heading of this section was added into the new revision) in which the HAPF results for the modified algorithm were compared to the HAPF operation without such modifications. Also section 4 shows how the proposed solution affect the effectiveness of the HAPF.
Comment 2: If the proposal has been tested in real mining conditions, this should be mentioned and illustrated by photos (experiments context).
Answer: This is a linguistic error in the abstract that has been corrected in this revision. By 'real' we meant 'not simulations'. The experiments were carried out in conditions corresponding to the real ones, but performed in the laboratory. We would like to add that our solution is currently in the implementation phase and such detailed measurements would not be possible in real industrial conditions due to the explosion-proof protection.
Reviewer 2 Report
1.In the sentence redundant term is found where the meaning is unclear or ambiguous.
2. lot of grammatical mistakes even in the abstract.
3. technical definition or evidence of the contribution of the work developed explain clearly.
4. Author did not mention their exact contribution.
5.figure 2 need more detail explanation.
6.experimental section need more detail explanation each parameters used in experiments.
Author Response
The authors would like to thank the Reviewer for the effort in reviewing the paper.
Comment 1: In the sentence redundant term is found where the meaning is unclear or ambiguous.
Answer: Because this comment does not indicate the specific sentence we do not respond to it. However we want to inform that the English language proofreading of the paper has been done and grammar or style mistakes have been corrected.
Comment 2: Lot of grammatical mistakes even in the abstract
Answer: The English language proofreading of the paper has been done.
Comment 3: technical definition or evidence of the contribution of the work developed explain clearly
and
Comment 4: Author did not mention their exact contribution
Answer: In the opinion of the authors, it was explained in the article, both in the introduction and in the summary. An original solution related to the control strategy was proposed. Based on known method with the feedback and feedforward control the following improvements have been added:
- compensation of the influence of distortions in the network voltages
- compensation of processing delays
- time critical implementation of the control algorithm
In addition the appropriate converter topology with SiC transistors allowing to achieve high energy efficiency has been proposed.
Comment 5: Figure 2 need more detail explanation.
Answer: This figure has been supplemented with additional details. Now, the detailed description of individual parts can be found in the content of the paper.
Comment 6: Experimental section need more detail explanation each parameters used in experiments.
Answer: Experimental section has been supplemented with additional details (e.g. table 2). Moreover, this section was reorganized by adding subsections.
Reviewer 3 Report
- Terms "power quality", "hybrid active power filters", "control methods", "time delay effect", and "explosion hazard zone" are not used in abstract;
- What is the purpose to implement the SiC transistor in this paper? The switching frequency is 20kHz and usually super junction MOSFET even IGBT can be used in this application. There is no size limitation in HAPF so using several MOSFETs in parallel can also reduce the Rds(on);
- There is no current loop in proposed method. How to maintain the system stability during load transient?
- The PI design for voltage loop should be discussed in order to verify the system dynamic response;
- What are transfer functions of HPF and LPF in CL1 and CL2, respectively? Are there any design considerations for them by proposed method?
- How to decide K factor in CL1?
Author Response
The authors would like to thank the Reviewer for the effort in reviewing the paper. We also want to inform that the English language proofreading of the paper has been done.
Comment 1: Terms "power quality", "hybrid active power filters", "control methods", "time delay effect", and "explosion hazard zone" are not used in abstract;
Answer: Both keywords and the abstract have been corrected
Comment 2: What is the purpose to implement the SiC transistor in this paper? The switching frequency is 20kHz and usually super junction MOSFET even IGBT can be used in this application. There is no size limitation in HAPF so using several MOSFETs in parallel can also reduce the Rds(on);
Answer: At the design stage of the hybrid active power filter one of the most important aspect was the limitation of voltage drop effects on converter transistors. Our previous experience with IGBT converters applied to HAPF had shown that threshold VCE voltages in IGBTs generate unwanted harmonics in output voltages and causes harmonic currents which are hard to be eliminated. Because MOSFETs do not have threshold voltage were found perfect solution to these issues. In our laboratory we had also tremendous issues with hard switched Si-MOSFETs like CoolMOS, do to relatively high reverse recovering charge of their body diodes. Therefore the choice of SiC MOSFET for HAPF seemed to be the best one. Even the switching frequency of SiC MOSFET could be higher than the chosen 20 kHz, this limit has been selected due to power losses.
Comment 3: There is no current loop in proposed method. How to maintain the system stability during load transient?
Answer: Indeed, there is no current loop in the proposed solution because the controlled variables are the output voltages. Stability is an important aspect in the case of CL1 which is based on the feedback loop. Stability in load transients is influenced by the parameters of the passive filter, the time delay, the gain K and the LPF and HPF filters. Therefore, section 5.3 was added to the new revision. This section deals with the stability of the control system.
Comment 4: The PI design for voltage loop should be discussed in order to verify the system dynamic response;
Answer: Section 5.1, which describes the selection of PI controller parameters, has been added to the new revision.
Comment 5: What are transfer functions of HPF and LPF in CL1 and CL2, respectively? Are there any design considerations for them by proposed method?
Answer: Yes, there are design considerations for transfer functions of HPF and LPF. These are the signal low pass and high pass filters and their parameters affect the dynamics, the effectiveness of higher harmonics reduction as well as the stability (HPF in CL1). Section 5.2 has been added into the new revision, which briefly describes the selection of LPF and HPF parameters.
Comment 6: How to decide K factor in CL1?
Answer: As already mentioned, the effectiveness of the reduction of higher harmonics depends on the K factor, but it also affects the stability. The analysis of the selection of this factor is shown in the added subsection 5.3.
Round 2
Reviewer 3 Report
- The term "time delay effect" is not shown in the abstract;
- Voltage drop effects on converter transistors is not important with HAPF since the dc-link voltage is high. If this is considered as conversion efficiency, this portion should be discussed.
Round 3
Reviewer 3 Report
The switching loss is not important, either since the power rating of HAPF is high (rated line-to-line voltage RMS value is 1000 V and rated phase current RMS value is 30 A).
